# Discovery of Novel Triazole-Containing Pyrazole Ester Derivatives as Potential Antibacterial Agents

**DOI:** 10.3390/molecules24071311

**Published:** 2019-04-03

**Authors:** Ming-Jie Chu, Wei Wang, Zi-Li Ren, Hao Liu, Xiang Cheng, Kai Mo, Li Wang, Feng Tang, Xian-Hai Lv

**Affiliations:** 1School of Science, Anhui Agricultural University, Hefei 230036, China; chumingjie@ahau.edu.cn (M.-J.C.); weiwang133@163.com (W.W.); renzilix@163.com (Z.-L.R.); 13167720906@163.com (H.L.); xiangcheng215b@163.com (X.C.); 18256549679@163.com (K.M.); Wangli-hx@ahau.edu.cn (L.W.); 2International Center for Bamboo and Rattan, 8 Fu Tong East Street, Beijing 100714, China; fengtang@icbr.ac.cn; 3Division of Chemistry and Biological Chemistry, School of Physical and Mathematical Sciences, Nanyang Technological University, Singapore 637371, Singapore

**Keywords:** pyrazole, triazole, ester, antibacterial, topoisomerase II inhibitor

## Abstract

To develop new antibacterial agents, a series of novel triazole-containing pyrazole ester derivatives were designed and synthesized and their biological activities were evaluated as potential topoisomerase II inhibitors. Compound **4d** exhibited the most potent antibacterial activity with Minimum inhibitory concentration (MIC) alues of 4 µg/mL, 2 µg/mL, 4 µg/mL, and 0.5 µg/mL against *Staphylococcus aureus*, *Listeria monocytogenes*, *Escherichia coli*, and *Salmonella gallinarum*, respectively. The in vivo enzyme inhibition assay **4d** displayed the most potent topoisomerase II (IC_50_ = 13.5 µg/mL) and topoisomerase IV (IC_50_ = 24.2 µg/mL) inhibitory activity. Molecular docking was performed to position compound **4d** into the topoisomerase II active site to determine the probable binding conformation. In summary, compound **4d** may serve as potential topoisomerase II inhibitor.

## 1. Introduction

Bacterial type II topoisomerases (DNA gyrase and topoisomerase IV) are ubiquitous and essential enzymes that have critical roles in the fundamental biological processes of replication, transcription, recombination, repair, and chromatin remodeling [1,2]. The action of topoisomerases is to change the spatial structure of DNA by catenation and decatenation of duplex DNA rings, relaxation of supercoiled DNA, and in the use of DNA topoisomerase II introduction of negative supercoils into DNA in an energy-dependent reaction [3]. Therefore, bacterial DNA topoisomerase II has drawn much attention as a selected target for finding potent antibacterial agents [4,5,6]. For example, a lot of synthetic quinolone antibacterial agents have been marketed and are now widely used for the treatment of bacterial infectious diseases [7,8]. Specifically, sparfloxacin has proven to be a very successful inhibitor of bacterial DNA topoisomerase II [9]. However, the abuse of antibacterial has led to an increase in bacterial resistance, especially in immunocompromised patients [10]. Our current research efforts are to find novel antibacterial agents as DNA topoisomerase II inhibitors.

Hybridization was a common method of drug development and design [11]. It was based on combining two or more different biologically active moieties in a single molecule to obtain the corresponding conjugated hybrid molecules [12]. These hybrid molecules may exhibit better activities than their precursors [13]. The novel synthesized heterocyclic molecules can work by the same or different mechanisms of action compared to the precursors [14]. Nitrogen containing heterocyclic may be potential antibacterial drugs which inhibit bacterial topoisomerases [15], such as triazoles [16], quinolones [17], oxazolopyridines, aminopyrazinamides, and pyrazole [18,19,20]. Recently, Plech [21] and Lal [22] reported triazole derivatives as potential topoisomerase II inhibitors, which exhibited greater antibacterial activities than ciprofloxacin (Figure 1).

As is well known, prolonged use of antibacterial agents can lead to environmental problems and residual toxicity [23,24]. However, compound containing ester groups, can easily be hydrolyzed to produce low toxicity compounds. In addition, it can increase its soluble fat in vivo and enhance antimicrobial activity. Furthermore, it has multiple charges, enabling them to interact with the bacterial cell surface more strongly than their monomeric counterparts. Recently, Haldar et al. [25] reported that two compounds were linked by an ester group, and the antibacterial activities of the new compounds were improved compared with the previous compounds. In the previous research work [8], we discovered a novel Topoisomerase II (Topo II) inhibitors compound 3, which showed considerable inhibitory activity than ciprofloxacin. In the search for a class of new Topoisomerase II inhibitors, we chose pyrazole as primer molecule, introduced a triazole structure, and used the ester group as a linker to obtain a new scaffold (Figure 2) [26,27]. After that, a series of novel triazole-containing pyrazole ester derivatives were designed and synthesized. Initially, all of the synthesized compounds were used against two fungi, but the results of the antifungal activity were not ideal. Therefore, four bacteria were used to test the antibacterial activity of these compounds, two Gram-positive bacteria, namely *Staphylococcus aureus* and *Listeria monocytogenes*, and two Gram-negative bacteria, namely *Escherichia coli* and *Salmonella gallinarum.* In addition, molecular docking analysis studies were performed on all derivatives to determine critical structural factors responsible for their antibacterial efficacy.

## 2. Results and Discussion

### 2.1. Chemistry

A series of triazole-containing pyrazole ester derivatives were prepared and the route listed in Scheme 1. We dissolved substituted phenylhydrazine with ethanol, then added ethylacetoacetate. Preparation of **1a**–**d** was with Vilsmeier–Haack Reagent (*N*,*N*-Dimethylformamide/POCl_3_). Then, we added KMnO_4_ as an oxidation reagent to obtain **2a**–**d**. Compounds **3a**–**d** were obtained from phenylhydrazine hydrochloride and urea, we used formic acid under acidic conditions. The preparation of the target compounds was carried out using the reported method [28]. We analyzed all prepared compounds using spectral and elemental methods, which showed that all compounds fully complied with the structure in Table 1.

### 2.2. Antibacterial Activity and Structure-Activity Relationships (SAR) Discussion

In vitro antimicrobial activity of the prepared compounds against four bacteria was evaluated by the conventional agar-dilution method. Ciprofloxacin was selected as a reference standard. The results of the in-vitro antibacterial activity screening of the test compounds are summarized in Table 1.

The results revealed that most of the synthetic compounds exhibited antibacterial activities *Staphylococcus aureus*, *Listeria monocytogenes*, *Escherichia coli,* and *Salmonella gallinarum*, demonstrating the rationality of our design strategy. As can be seen from Table 1, among all the intermediate compounds, compound 2d exhibited the most potent antibacterial activity with MIC values of 16 µg/mL against *Staphylococcus aureus*, but among the tested target compounds, three compounds (i.e., **4d**, **4g**, and **4k**) were found to display improved antibacterial activities against *Staphylococcus aureus* compared with (16 mg/L), with MIC values ranging from 4 to 8 mg/L. The eye-catching finding was that the target products greater excellent antibacterial activity than intermediate compounds. Among them, compounds **4d** and **4k** displayed most potent activity with MIC values of 4, 2, 4, and 0.5 µg/mL and 4, 8, 4, and 4 µg/mL against *Staphylococcus aureus*, *Listeria monocytogenes*, *Escherichia coli,* and *Salmonella gallinarum*, which were similar to the broad-spectrum antibiotic Ciprofloxacin), indicating that they possess antibacterial activity. Compound **4a** showed antibacterial activities against *Staphylococcus aureus*, *Listeria monocytogenes*, *Escherichia coli,* and *Salmonella gallinarum*, Then, we introduced different substituent groups on benzene ring in **4a**, the introduction of methyl at **R^1^** displayed significantly enhanced effects compared with that of other compounds. Compound **4d** exhibited best antibacterial activity than others. When **R**^2^ position was substituted with fluorine, methyl substituted at the **R**^1^ site was beneficial to antibacterial activity of the synthesized compounds (e.g., **4h**). When the **R**^2^ position was substituted with methyl, electron-donating substituents (e.g., **4k**) at **R**^1^ showed more potent activities than those with electron-withdrawing (e.g., **4l**, **4m**). When **R**^1^ and **R**^2^ were simultaneously substituted by electron-withdrawing, their activities were poor compared with other compounds. (e.g., **4j**, **4f**, **4i**). Considering the above SAR results, these findings suggest that the antibacterial potency of designed compounds could be ascribed to a combination of factors, such as the **R^1^** position was substituted with electron-donating groups, which may enhance antibacterial activity, and that the **R**^2^ position was substituted with hydrogen and the compound may enhance antibacterial activity.

### 2.3. Inhibitory Effects against DNA Gyrase and Topoisomerase IV

In order to determine the relationship between compounds and antibacterial activity, the inhibitory activity of selected compounds (**4d** and **4k**) against DNA gyrase and topoisomerase IV isolated from *Escherichia coli* was examined. As shown in Table 2, **4d** showed more potent inhibition than **4k** against the two enzymes. The same exhibited the same tendency as the MIC data, suggesting that compound **4d** may serve as a potential topoisomerase II inhibitor.

### 2.4. Docking Analysis

In order to gain a better understanding on the potency of the synthesized compounds and guide further structure, we conducted activity relationships studies. All of the derivatives were docked into the active site of Topo II (PDB entry: 2xcs) in the binding model of compound **4d** and Topo II. The skeleton of compound **4d** was embedded in the binding pocket, showing that the pose of **4d** into the Topo II-binding site had a suitable shape that was complementary to the binding pocket, which means that our design strategy was rational. E:Dc11, F:Dc11, and F:Dg10 established π–π interactions with a benzene ring of compound **4d** (distance: 5.01 Å, 4.18 Å, 5.31 Å), E:Dc11 established π–π interactions with triazole ring of compound **4d** (distance: 4.60 Å), Arg 1122 established cation–π interactions with pyrazole ring of compound **4d** (distance: 5.26 Å) (Figure 3).

## 3. Experimental Section

### 3.1. Materials and Methods

All chemicals were purchased from Energy, Meryer, and Aladdin Chemicals and were used as received. ^1^H-NMR spectra analyses were carried out using anAgilent DD2 600 Hz spectrometer with CDCl_3_ as the solvent and tetramethyllsilane as the internal standard. ESI-MS (Electrospray Ionisation Mass Spectrometry) spectra were carried out on a Mariner System 5304 mass spectrometer. Elemental analyses were performed on a CHN–O–Rapid instrument. Molecular docking was performed with Discovery Studio 3.5 [29]. The ^13^C-NMR and ESI-MS spectrum of compounds can be found in the Appendix A.

#### 3.1.1. General Procedure for the Synthesis of 5-Chloro-1-aryl-3-methyl-1*H*-pyrazole-4-carboxylic Acids **2a**–**d**

Intermediates **2a**–**d** were obtained from Reference [30]. Dissolve para-substituted phenyl hydrazine (0.025 mol) with anhydrous ethanol ethyl acetoacetate (0.025 mol) was added portion wise, stirred, and refluxed for 5 h, then the solution was rotary evaporated to form a solid, which was dissolved in DMF (25 mL) and phosphorus oxychloride (20 mL) of cold mixed solution and stirred at 85 °C for 2 h. The resulting product was poured into ice-cold water and the solid was isolated by filtration to give a yellow solid, which was oxidized by KMnO_4_ solution and stirred at 70–80 °C. After cooling to room temperature, the pH was adjusted to alkaline by the addition of 10% NaOH solution, and the solution was filtered, HCl solution was added to the solution and solid **2a**–**d** eventually separated out. The resulting crude product was recrystallized from anhydrous ethanol to give the pure product.

#### 3.1.2. General Pathway for Prepare of 1-phenyl-1*H*-1,2,4-triazol-3-ol **3a**–**d**

Intermediates **3a**–**d** were synthesized according to Bin sun et al. [31]. Phenylhydrazine (0.2 mol) and urea (0.25 mol) were dissolved in water (50 mL) and 30% hydrochloric acid (64.8 g) was added for 4 h at 135 °C. Add formic acid (0.3 mol) and 98% concentrated sulfuric acid (4.8 g) at 90 °C for 6 h. Finally, the product was cooled to room temperature, filtered, and washed with water to neutral and dried to give solid **3a**–**d**.

#### 3.1.3. General Procedure for Synthesis of 1-phenyl-1*H*-1,2,4-triazol-3-yl 5-chloro-3-methyl-1-phenyl-1*H*-pyrazole-4-carboxylate **4a**–**m**

Compounds **2a**–**d** (1 mmol) was stirred with triethylamine (1.5 mmol) into DMF (10 mL) medium, then a mixture of DCC (1 mmol) and DMAP (1 mmol) was added in the reaction system, stirred at 25 °C for 1 h. The mixture of intermediate **3** (1.2 mmol) and DMF (6 mL) was added in the reaction, stirred at 25 °C for 3 h. Finally, the product was extracted from chloroform with water, hydrochloric acid (0.2 mol/L), sodium hydroxide (2 mol/L), saturated sodium chloride successively, and then dried, concentrated, and purified by preparative thin layer chromatography (PE: EA = 8:1) [27]. 

#### 3.1.4. 1-phenyl-1*H*-1,2,4-triazol-3-yl5-chloro-3-methyl-1-phenyl-1*H*-pyrazole-4-carboxylate (**4a**)

White solid, yield 62%; mp 138–140 °C; ^1^H-NMR (600 MHz, CDCl_3_) δ 8.00 (s, 1H, Triazole-H), 7.53 (s, 6H, Ph-H), 7.08 (d, *J* = 8.0 Hz, 1H, Ph-H), 6.98 (t, *J* = 8.6 Hz, 1H, Ph-H), 6.93 (d, *J* = 4.4 Hz, 1H, Ph-H), 6.88 (d, *J* = 8.2 Hz, 1H, Ph-H), 2.55 (s, 3H, CH_3_). MS (ESI) calculated for C_19_H_15_ClN_5_O_2_ [M + H]^+^, 380.1, found 380.1; Anal. Calcd for C_19_H_14_ClN_5_O_2_: C, 60.09; H, 3.72; N, 18.44%; Found: C, 60.12; H, 3.79; N, 18.33%.

#### 3.1.5. 1-phenyl-1*H*-1,2,4-triazol-3-yl5-chloro-1-(4-fluorophenyl)-3-methyl-1*H*-pyrazole-4-carboxylate (**4b**)

White solid, yield 54%; mp166–168 °C; ^1^H-NMR (600 MHz, CDCl_3_) δ 7.95 (s, 1H, Triazole-H), 7.51 (dd, *J* = 8.8, 4.7 Hz, 2H, Ph-H), 7.28 (s, 1H, Ph-H), 7.25 (s, 1H, Ph-H), 7.22 (t, *J* = 8.4 Hz, 2H, Ph-H), 6.98–6.91 (m, 3H, Ph-H), 2.54 (s, 3H, CH_3_).MS (ESI) calculated for C_19_H_14_ClFN_5_O_2_ [M + H]^+^, 398.1, found 398.1; Anal. Calcd for C_19_H_13_ClFN_5_O_2_: C, 57.37; H, 3.29; N, 17.61%; Found: C, 57.41; H, 3.31; N, 17.73%.

#### 3.1.6. 1-phenyl-1*H*-1,2,4-triazol-3-yl5-chloro-1-(4-chlorophenyl)-3-methyl-1*H*-pyrazole-4-carboxylate (**4c**)

White solid, yield 57%; mp 172–173 °C; ^1^H-NMR (600 MHz, CDCl_3_) δ 7.92 (s, 1H, Triazole-H), 7.49 (d, *J* = 2.3 Hz, 4H, Ph-H), 7.28 (s, 1H, Ph-H), 7.25 (s, 1H, Ph-H), 6.98–6.90 (m, 3H, Ph-H), 2.54 (s, 3H, CH_3_) MS (ESI) calculated for C_19_H_14_Cl_2_N_5_O_2_ [M + H]^+^,414.1,found 414.1; Anal. Calcd for C_19_H_13_Cl_2_N_5_O_2_: C, 55.09; H, 3.16; N, 16.91%; Found: C, 55.13; H, 3.24; N, 17.05%.

#### 3.1.7. 1-phenyl-1*H*-1,2,4-triazol-3-yl5-chloro-3-methyl-1-(p-tolyl)-1*H*-pyrazole-4-carboxylate (**4d**)

White solid, yield 52%; mp 154–156 °C; ^1^H-NMR (600 MHz, CDCl_3_) δ8.42 (s, 1H, Triazole-H), 7.57 (d, *J* = 8.4 Hz, 6H, Ph-H), 7.49 (d, *J* = 8.3 Hz, 1H, Ph-H), 7.31 (t, *J* = 8.2 Hz, 2H, Ph-H), 2.62 (s, 3H, CH_3_), 2.42 (s, 3H, CH_3_). [M + H]^+^). MS (ESI) calculated for C_20_H_17_ClN_5_O_2_ [M + H]^+^, 394.1, found 394.1; Anal. Calcd for C_20_H_16_ClN_5_O_2_: C, 61.00; H, 4.10; N, 17.78%; Found: C, 61.05; H, 4.18; N, 17.68%.

#### 3.1.8. 1-(4-fluorophenyl)-1*H*-1,2,4-triazol-3-yl5-chloro-3-methyl-1-phenyl-1*H*-pyrazole-4-carboxylate (**4e**)

White solid, yield 47%; mp 187–189 °C; ^1^H-NMR (600 MHz, CDCl_3_) δ7.97 (s, 1H, Triazole-H), 7.58–7.43 (m, 6H, Ph-H), 6.98–6.91 (m, 3H, Ph-H), 2.55 (s, 3H, CH_3_). MS (ESI) calculated for C_19_H_14_ClFN_5_O_2_ [M + H]^+^, 398.1, found 398.1; Anal. Calcd for C_19_H_13_ClFN_5_O_2_: C, 57.37; H, 3.29; N, 17.61%; Found: C, 57.43; H, 3.30; N, 17.57%.

#### 3.1.9. 1-(4-fluorophenyl)-1*H*-1,2,4-triazol-3-yl5-chloro-1-(4-fluorophenyl)-3-methyl-1*H*-pyrazole-4-carboxylate (**4f**)

White solid, yield 75%; mp 69–71 °C; ^1^H-NMR (600 MHz, CDCl_3_) δ 8.03 (s, 1H, Triazole-H), 7.89 (d, *J* = 8.2 Hz, 2H, Ph-H), 7.51 (dd, *J* = 8.8, 4.7 Hz, 2H, Ph-H), 7.35 (d, *J* = 8.1 Hz, 2H, Ph-H), 7.20 (d, *J* = 8.2 Hz, 2H, Ph-H), 2.54 (s, 3H, CH_3_). MS (ESI) calculated for C_19_H_13_ClF_2_N_5_O_2_ [M + H]^+^, 416.1, found 416.1; Anal. Calcd for C_19_H_12_ClF_2_N_5_O_2_: C, 54.89; H, 2.91; N, 16.84%; Found: C, 54.96; H, 3.10; N, 16.77%.

#### 3.1.10. 1-(4-fluorophenyl)-1*H*-1,2,4-triazol-3-yl5-chloro-1-(4-chlorophenyl)-3-methyl-1*H*-pyrazole-4-carboxylate (**4g**)

White solid, yield 66%; mp 253–254 °C; ^1^H-NMR (600 MHz, CDCl_3_) δ 8.43 (s, 1H, Triazole-H), 7.65 (dd, *J* = 8.2, 4.0 Hz, 2H, Ph-H), 7.54–7.48 (m, 4H, Ph-H), 7.21 (t, *J* = 8.2 Hz, 2H, Ph-H), 2.61 (s, 3H, CH_3_). MS (ESI) calculated for C_19_H_13_Cl_2_FN_5_O_2_ [M+H]^+^, 432.0, found 432.0; Anal. Calcd for C_19_H_12_Cl_2_FN_5_O_2_: C, 52.80; H, 2.80; N, 16.20%; Found: C, 52.87; H, 2.77; N, 16.28%.

#### 3.1.11. 1-(4-fluorophenyl)-1*H*-1,2,4-triazol-3-yl5-chloro-3-methyl-1-(p-tolyl)-1*H*-pyrazole-4-carboxylate (**4h**)

White solid, yield 71%; mp 99–101 °C; ^1^H-NMR (600 MHz, CDCl_3_) δ 8.41 (s, 1H, Triazole-H), 7.66 (dd, *J* = 8.1, 4.5 Hz, 2H, Ph-H), 7.42 (d, *J* = 7.6 Hz, 2H, Ph-H), 7.31 (d, *J* = 7.9 Hz, 2H, Ph-H), 7.21 (t, *J* = 8.0 Hz, 2H, Ph-H), 2.60 (s, 3H, CH_3_), 2.43 (s, 3H. CH_3_). MS (ESI) calculated for C_20_H_16_ClFN_5_O_2_ [M + H]^+^, 412.1, found 412.1; Anal. Calcd for C_20_H_15_ClFN_5_O_2_: C, 58.33; H, 3.67; N, 17.01%; Found: C, 58.41; H, 3.59; N, 17.09%.

#### 3.1.12. 1-(4-chlorophenyl)-1*H*-1,2,4-triazol-3-yl5-chloro-1-(4-fluorophenyl)-3-methyl-1*H*-pyrazole-4-carboxylate (**4i**)

White solid, yield 51%; mp 84–86 °C; ^1^H-NMR (600 MHz, CDCl_3_) δ 8.46 (s, 1H, Triazole-H), 7.64 (d, *J* = 8.6 Hz, 2H, Ph-H), 7.55–7.54 (m, 2H, Ph-H), 7.50 (d, *J* = 8.6 Hz, 2H, Ph-H), 7.23–7.20 (m, 2H, Ph-H), 2.61 (s, 3H, CH_3_). MS (ESI) calculated for C_19_H_13_Cl_2_FN_5_O_2_ [M + H]^+^, 432.0, found 432.0; Anal. Calcd for C_19_H_12_Cl_2_FN_5_O_2_: C, 52.80; H, 2.80; N, 16.20%; Found: C, 52.86; H, 2.74; N, 16.11%.

#### 3.1.13. 1-(4-chlorophenyl)-1*H*-1,2,4-triazol-3-yl5-chloro-1-(4-chlorophenyl)-3-methyl-1*H*-pyrazole-4-carboxylate (**4j**)

White solid, yield 56%; mp 72–74 °C; ^1^H-NMR (600 MHz, CDCl_3_) δ 8.42 (s, 1H, Triazole-H), 7.55–7.53 (d, *J* = 3.1 Hz, 4H, Ph-H), 7.31 (dd, *J* = 8.9, 4.7 Hz, 2H, Ph-H), 7.20 (t, *J* = 8.4 Hz, 2H, Ph-H), 2.42 (s, 3H, CH_3_). MS (ESI) calculated for C_19_H_13_Cl_3_N_5_O_2_ [M + H]^+^, 448.0, found 448.0; Anal. Calcd for C_19_H_12_Cl_3_N_5_O_2_: C, 50.86; H, 2.70; N, 15.61%; Found: C, 50.91; H, 2.64; N, 15.58%.

#### 3.1.14. 1-(p-tolyl)-1*H*-1,2,4-triazol-3-yl-5-chloro-3-methyl-1-phenyl-1*H*-pyrazole-4-carboxylate (**4k**)

White solid, yield 72%; mp 135–138 °C; ^1^H-NMR (600 MHz, CDCl_3_) δ 8.43 (s, 1H, Triazole-H), 7.57–7.46 (m, 7H, Ph-H), 7.31-7.40 (d, *J* = 8.2 Hz, 2H, Ph-H), 2.62 (s, 3H, CH_3_), 2.42 (s, 3H, CH_3_).MS (ESI) calculated for C_20_H_17_ClN_5_O_2_ [M + H]^+^, 394.1, found 394.1; Anal. Calcd for C_20_H_16_ClN_5_O_2_: C, 61.00; H, 4.10; N, 17.78%; Found: C, 61.08; H, 4.15; N, 17.84%.

#### 3.1.15. 1-(p-tolyl)-1*H*-1,2,4-triazol-3-yl5-chloro-1-(4-fluorophenyl)-3-methyl-1*H*-pyrazole-4-carboxylate (**4l**)

White solid, yield 46%; mp 141–143 °C; ^1^H-NMR (600 MHz, CDCl_3_) δ 8.43 (s, 1H, Triazole-H), 7.55–7.53 (m, 4H, Ph-H), 7.31 (d, *J* = 8.2 Hz, 2H, Ph-H), 7.23-7.21 (t, *J* = 8.5 Hz, 2H, Ph-H), 2.61 (s, 3H, CH_3_), 2.42 (s, 3H, CH_3_). MS (ESI) calculated for C_20_H_16_ClFN_5_O_2_ [M + H]^+^, 412.1, found 412.1; Anal. Calcd for C_20_H_15_ClFN_5_O_2_: C, 58.33; H, 3.67; N, 17.01%; Found: C, 58.42; H, 3.74; N, 17.07%.

#### 3.1.16. 1-(p-tolyl)-1*H*-1,2,4-triazol-3-yl5-chloro-1-(4-chlorophenyl)-3-methyl-1*H*-pyrazole-4-carboxylate (**4m**)

White solid, yield 49%; mp 174–176 °C; ^1^H-NMR (600 MHz) δ 7.97 (s, 1H, Triazole-H), 7.53 (d, *J* = 4.2 Hz, 4H, Ph-H), 7.49 (dd, *J* = 8.9, 4.5 Hz, 1H, Ph-H), 7.00–6.91 (m, 3H, Ph-H), 2.55 (s, 3H, CH_3_). MS (ESI) calculated for C_20_H_16_Cl_2_N_5_O_2_ [M + H]^+^, 428.1, found 428.1; Anal. Calcd for C_20_H_15_Cl_2_N_5_O_2_: C, 56.09; H, 3.53; N, 16.35%; Found: C, 56.13; H, 3.61; N, 16.44%.

### 3.2. In Vitro Antibacterial Activity

#### 3.2.1. Medium

The solid media Luria-Bertani (LB)-Broth-Agar-Medium (tryptone 10 g/L, yeast 5 g/L, NaCl 10 g/L, agar 15 g/L, and distilled water 1000 mL, adjusted to pH 7.4 was used for testing the antibacterial activity.

#### 3.2.2. Minimum Inhibitory Concentration (MIC)

The in vitro antibacterial activity for synthesized compounds **4a**–**m** were evaluated using the agar-dilution method [32,33,34]. Two-fold serial dilutions of the compounds and reference drugs (ciprofloxacin) was prepared in LB-Broth-Agar-Medium. Drugs (10.0 mg) were dissolved in DMSO (1 mL) and the solution was diluted with water (9 mL). Further progressive double dilution with melted LB-Broth-Agar-Medium was performed to obtain the required concentrations of 128, 64, 32, 16, 8, 4, 2, 1, 0.5 µg/mL, and the MIC values were calculated separately. The bacterial inocula were prepared by suspending 24 h-old bacterial colonies from LB-Broth-Agar-Medium in 0.85% saline. The inocula were adjusted to 0.5 McFarland Standard (1.56108 CFU/mL). The suspensions were then diluted in 0.85% saline to give 107 CFU/mL. Petri dishes were spot-inoculated with 1 μL of each of the prepared bacterial suspensions (104 CFU/spot) and incubated at 37 °C for 24 h. At the end of the incubation period, the MIC was determined, which is the lowest concentration of the test compound that resulted in no visible growth on the plate. A control test was also performed with test medium supplemented with DMSO at the same dilutions as used in the experiment in order to ensure that the solvent had no influence on bacterial growth.

### 3.3. Enzyme Inhibition Experimental

The in vitro antibacterial activity of the target compounds was carried out on by the methods of Sato et al. [35] and Peng and Marians [36]. First, the *E. coli* suspension is extracted to obtain a crude enzyme solution. After the purification step, the selected compounds are determined by gel electrophoresis to obtain data of different gradient concentrations, thereby calculating the IC_50_ values.

### 3.4. Molecular Docking

The crystal structures of bacterial DNA topoisomerase II (PDB entry: 2xcs) was downloaded from the RCSB (Research Collaboratory for Structural Bioinformatic) Protein Data Bank. The molecular docking procedure was performed using CDOCKER protocol for receptor–ligand interactions section of DS 3.5 [29].

## 4. Conclusions

A series of novel triazole-containing pyrazole ester derivatives **4a**–**m** were synthesized and evaluated for their biological activities. Compound **4d** showed significant antibacterial activities with MIC values of 4, 2, 4, and 0.5 µg/mL against *Staphylococcus aureus*, *Listeria monocytogenes*, *Escherichia coli*, and *Salmonella gallinarum.* Furthermore, compound **4d** displayed the most potent Topo II (IC_50_ = 13.5 µg/mL) and topoisomerase IV (IC_50_ = 24.2 µg/mL) inhibitory activity. This study suggests that compound **4d** can be used as a potential topoisomerase II inhibitor and this finding will lay the foundation for further structural modification and development of novel topoisomerase II inhibitors.

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
