# Peer review of "Discovery of Novel Triazole-Containing Pyrazole Ester Derivatives as Potential Antibacterial Agents"

_molecules, 2019, doi:10.3390/molecules24071311_

Reviewer 1 Report

Authors should cite the following :

Process for preparation of triazole containing pyrazole ester derivatives as bactericides
    By Lv, Xianhai; Cao, Haiqun; Ren, Zili; Chu, Mingjie
    From Faming Zhuanli Shenqing (2017), CN 107141283 A 20170908. 

Authors should mention it and state if there is anny additon in the article they present.

Author Response

Response to Reviewer 1 Comments

Prof. & Dr. Xian-Hai Lv, School of Science, Anhui Agricultural University, Hefei 230036, P.R. China

Tel./Fax: +86-551-6578-6906. E-mail: lvxianhai@ahau.edu.cn.

February 3th, 2019

Dear Reviewers:

Thank you for the reviewers’ comments concerning our manuscript entitled “ Discovery of novel triazole-containing pyrazole ester derivatives as potential antibacterial agents” (Manuscript ID: molecules-431506).Those comments are all valuable and very helpful for revising and improving our paper, as well as the important guiding significance to our researches. We have studied comments carefully and have made correction which we hope meet with approval. Revised portion are marked in red in the paper. The main corrections in the paper and the responds to the reviewer’s comments are as flowing:

Point 1: Process for preparation of triazole containing pyrazole ester derivatives as bactericides
    By Lv, Xianhai; Cao, Haiqun; Ren, Zili; Chu, Mingjie
    From Faming Zhuanli Shenqing (2017), CN 107141283 A 20170908.  

Authors should mention it and state if there is anny additon in the article they present.

Response: Thank you very much for reviewing our manuscript. According to your suggestion, we have already cited this patent in the Materials and methods section.

We appreciate for Reviewers’ warm work earnestly, and hope that the correction will meet with approval. Once again, thank you very much for your comments and suggestions.

Sincerely yours,

Xian-Hai Lv

Reviewer 2 Report

The manuscript entitled "Discovery of novel triazole-containing pyrazole ester derivatives as potential antibacterial agents" by Chu et al. presents synthesis, antibacterial activity and molecular docking new hybrid molecules.

The manuscript is not written in correct English and some sentences are unclear.

The "Chart 1" should be named "Scheme" or "Figure" and should contain the explanations of R1 and R2.

The section "Results and Discussion" contains in the text redundant MIC values, which are already listed in Table 1.

The discussion should contain a comparison of the presented activity data for hybrid compounds with activity of each hybrid components.

The spectral characteristic of newly synthesized compounds raises doubts.

There is lack of 1H NMR analysis, each of hydrogen atoms being at a specific position should be assigned to observed signal(s).

The results of MS (ESI) measurement for all compounds seem to be wrong, since the cited molecular masses correspond to calculated molecular masses, wherein the average molecular masses of elements were used for calculation.

I do not recommend the presented manuscript for publication in the Molecules.

Author Response

Response to Reviewer 2 Comments

Prof. & Dr. Xian-Hai Lv, School of Science, Anhui Agricultural University, Hefei 230036, P.R. China

Tel./Fax: +86-551-6578-6906. E-mail: lvxianhai@ahau.edu.cn.

February 3th, 2019

Dear Reviewers:

Thank you for the reviewers’ comments concerning our manuscript entitled “ Discovery of novel triazole-containing pyrazole ester derivatives as potential antibacterial agents” (Manuscript ID: molecules-431506).Those comments are all valuable and very helpful for revising and improving our paper, as well as the important guiding significance to our researches. We have studied comments carefully and have made correction which we hope meet with approval. Revised portion are marked in red in the paper. The main corrections in the paper and the responds to the reviewer’s comments are as flowing:

Point 1: The "Chart 1" should be named "Scheme" or "Figure" and should contain the explanations of R1 and R2.

Response: We are very grateful for your suggestions, we have made changes as you suggested and explained R1 and R2 in the Chemistry section.

Point 2: The section "Results and Discussion" contains in the text redundant MIC values, which are already listed in Table 1.

Response: Thanks for your suggestion, we have removed redundant MIC values.

Point 3: The discussion should contain a comparison of the presented activity data for hybrid compounds with activity of each hybrid components.

Response: Thank you very much for your suggestion, we added the experiment that in vitro antimicrobial activity of the intermediate 2a-2d and intermediate 3a-3d against four bacteria was evaluated by the conventional agar-dilution method. And add the results in Table 1. The results showed that the target products exhibited excellent antibacterial activity than intermediate.

Point 4: The spectral characteristic of newly synthesized compounds raises doubts. There is lack of 1H NMR analysis, each of hydrogen atoms being at a specific position should be assigned to observed signal(s).

Response: Thank you very much for reviewing our manuscript. We are sorry for our negligence. We have corrected it in the manuscript.

Point 5: The results of MS (ESI) measurement for all compounds seem to be wrong, since the cited molecular masses correspond to calculated molecular masses, wherein the average molecular masses of elements were used for calculation.

Response: We are sorry for our negligence. We have corrected it in the manuscript.

We appreciate for Reviewers’ warm work earnestly, and hope that the correction will meet with approval. Once again, thank you very much for your comments and suggestions.

Sincerely yours,

Xian-Hai Lv

Reviewer 3 Report

This paper describes synthesis and biological evaluation of novel triazol-containing pyrazole derivertives. According to following reasons, this manuscript can not be recommended for publication in Molecules as it stands. 

1)      In the introduction, the authors review importance of DNA topoisomerase inhibitors to develop potential antibacterial agents. In addition, the authors emphasize the importance of hybridization methods in drug discovery. However, the relations between the present and the previous works are not clearly described. The authors should describe more clearly the concept for design of the synthesized compounds.

2)      For example, the authors should describe theoretical importance for triazol and pyrazole scaffolds in your design. Accordingly Figure 2 should be revised. Moreover, it is unclear why two fragments are directly linked by the ester linker. Is the length of the linker optimized by the authors?  .

3) Biological activities and toxicity of two fragments, which constitute the hybridized compounds, should be examined and compared with those of hybridized compounds. These experimental would be very important to support the author’s hypothesis regarding environmental problems

Author Response

Response to Reviewer 3 Comments

Prof. & Dr. Xian-Hai Lv, School of Science, Anhui Agricultural University, Hefei 230036, P.R. China

Tel./Fax: +86-551-6578-6906. E-mail: lvxianhai@ahau.edu.cn.

February 3th, 2019

Dear Reviewers:

Thank you for the reviewers’ comments concerning our manuscript entitled “ Discovery of novel triazole-containing pyrazole ester derivatives as potential antibacterial agents” (Manuscript ID: molecules-431506).Those comments are all valuable and very helpful for revising and improving our paper, as well as the important guiding significance to our researches. We have studied comments carefully and have made correction which we hope meet with approval. Revised portion are marked in red in the paper. The main corrections in the paper and the responds to the reviewer’s comments are as flowing:

Point 1: In the introduction, the authors review importance of DNA topoisomerase inhibitors to develop potential antibacterial agents. In addition, the authors emphasize the importance of hybridization methods in drug discovery. However, the relations between the present and the previous works are not clearly described. The authors should describe more clearly the concept for design of the synthesized compounds.

Response: We are grateful for your in-depth review. We have made some brief descriptions in the introduction section as per your suggestion

Point 2: For example, the authors should describe theoretical importance for triazol and pyrazole scaffolds in your design. Accordingly Figure 2 should be revised. Moreover, it is unclear why two fragments are directly linked by the ester linker. Is the length of the linker optimized by the authors? .

Response: We are very grateful for your suggestions. We have made some changes in the Figure 2. The compound containing ester group, which can be easily hydrolyzed to produce low toxicity compound. And it can increase its fat-soluble in vivo and enhance antimicrobial activity. So we used ester groups as a linker and expected better biological activity.

Point 3: Biological activities and toxicity of two fragments, which constitute the hybridized compounds, should be examined and compared with those of hybridized compounds. These experimental would be very important to support the author’s hypothesis regarding environmental problems

Response: Thank you very much for your suggestion, we added the experiment that in vitro antimicrobial activity of the intermediate 2a-2d and intermediate 3a-3d against four bacteria was evaluated by the conventional agar-dilution method. And add the results in Table 1. The results showed that the target products exhibited excellent antibacterial activity than intermediate.

We appreciate for Reviewers’ warm work earnestly, and hope that the correction will meet with approval. Once again, thank you very much for your comments and suggestions.

Sincerely yours,

Xian-Hai Lv

Round  2

Reviewer 2 Report

English has not been improved, there are still many basic mistakes.

The discussion of the results should be more comprehensive. The Authors have only added MIC values to Table 1, without sufficient comments in the text.

1H NMR analysis is still not satisfying. The individual hydrogen atoms in the aromatic rings should be assigned to the specific signals.

The results of MS (ESI) measurement seem doubtful. The Authors are requested to submit MS spectra for verification.

Author Response

Response to Reviewer 2 Comments

Prof. & Dr. Xian-Hai Lv, School of Science, Anhui Agricultural University, Hefei 230036, P.R. China

Tel./Fax: +86-551-6578-6906. E-mail: lvxianhai@ahau.edu.cn.

February 10th, 2019

Dear Reviewers:

Thank you for the reviewers’ comments concerning our manuscript entitled “ Discovery of novel triazole-containing pyrazole ester derivatives as potential antibacterial agents” (Manuscript ID: molecules-431506).Those comments are all valuable and very helpful for revising and improving our paper, as well as the important guiding significance to our researches. We have studied comments carefully and have made correction which we hope meet with approval. Revised portions are marked in red in the paper. The main corrections in the paper and the responds to the reviewer’s comments are as flowing:

Point 1: The discussion of the results should be more comprehensive. The Authors have only added MIC values to Table 1, without sufficient comments in the text.

Response: We are grateful for your in-depth review. According to your suggestion, we have made some brief descriptions in Antibacterial activity and SAR Discussion section as per your suggestion.

Point 2: 1H NMR analysis is still not satisfying. The individual hydrogen atoms in the aromatic rings should be assigned to the specific signals.

Response: Thank you very much for your careful review. Two phenyl-H signals are very similar, the difference is small, even some signals overlap. I hope you understand that it is difficult for us to make an accurate judgment.

Point 3: The results of MS (ESI) measurement seem doubtful. The Authors are requested to submit MS spectra for verification.

Response: We are embarrassed by this kind of mistake, thank you for your review. We have been re-analyzed their data and corrected this systematic mistake according to the Reviewer’s suggestion.

We appreciate for Reviewers’ warm work earnestly and hope that the correction will meet with approval. Once again, thank you very much for your comments and suggestions.

Sincerely yours,

Xian-Hai Lv

Reviewer 3 Report

The revised paper by Lv describes synthesis and microbiological activity of triazole-containing pyrazole ester derivatives. The manuscript has been revised significantly according to comments suggested by the previous referees. Therefore this reviewer recommends the manuscript be published in Molecules.

Author Response

Response to Reviewer 3 Comments

Prof. & Dr. Xian-Hai Lv, School of Science, Anhui Agricultural University, Hefei 230036, P.R. China

Tel./Fax: +86-551-6578-6906. E-mail: lvxianhai@ahau.edu.cn.

February 10th, 2019

Dear Reviewers:

We appreciate for Reviewers’ warm work earnestly. Once again, thank you very much for your comments and suggestions.

Sincerely yours,

Xian-Hai Lv